# Antibiotic dispensing behaviors for pediatric diarrhea: A qualitative study of informal healthcare providers in rural Bangladesh

Debashish Biswas[1,2*], Melissa H. Watt[3‡], Aparna Mangadu[4], Jyoti Bhushan Das[1], Mohammad Saeed Munim[1], Ridwan Mostofa Shihab[1], Zahid Hasan Khan[5], Mohammad Ashraful Amin[5], Ishtiakul Islam Khan[5], Md. Taufiqul Islam[5], Olivia R. Hanson[4], Eric J. Nelson[6], Firdausi Qadri[5], Daniel Leung[4‡], Ashraful Islam Khan[5*]

1 Health Systems and Population Studies Division, International Centre for Diarrhoeal Disease Research, Bangladesh (icddr,b), Dhaka, Bangladesh, 2 School of Population and Global Health, The University of Western Australia, Perth, Western Australia, Australia, 3 Department of Population Health Sciences, Spencer Fox Eccles School of Medicine at the University of Utah, Salt Lake City, Utah, United States of America, 4 Division of Infectious Diseases, Department of Internal Medicine, Spencer Fox Eccles School of Medicine at the University of Utah, Salt Lake City, Utah, United States of America, 5 Infectious Diseases Division, International Centre for Diarrhoeal Disease Research, Bangladesh (icddr,b), Dhaka, Bangladesh, 6 Department of Pediatrics, University of Florida, Gainesville, Florida, United States of America,

‡ Co-senior authors.
* debashish@icddrb.org (DB); ashrafk@icddrb.org (AIK)

## Abstract

Antibiotic resistance is a significant public health concern and requires coordinated efforts to promote antibiotic stewardship. In rural Bangladesh, informal healthcare providers known as "village doctors" are a primary source of antibiotics for a range of ailments, including pediatric diarrhea. This qualitative study explored the factors influencing antibiotic dispensing practices for pediatric diarrhea among village doctors in the Chattogram district of Bangladesh. In May 2023, we conducted in-depth interviews with 18 village doctors and analyzed the data using thematic analysis approach. The resulting themes were mapped onto the five domains of the Social Ecological Framework (SEF) to identify the multi-level drivers of antibiotic dispensing practices. At the individual level, antibiotic provision for treatment of pediatric diarrhea is influenced by village doctors' clinical knowledge, training, and beliefs about antibiotics. At the interpersonal level, village doctors often alter their antibiotic dispensing practices based on patient and caregiver demands and influences from pharmaceutical companies. Organizational level factors include financial incentives from medication sales and pharmaceutical companies, resource constraints, diagnostic limitations, and a competitive healthcare landscape. We also identified that cultural expectations for rapid recovery through antibiotics and widespread community access to antibiotics at the community level. Finally, at the policy level, we identified weak regulatory frameworks, inadequate enforcement mechanisms, and healthcare system challenges that position village doctors as essential but unregulated

**Data availability statement:** Given the difficulty of fully de-identifying qualitative data, the research team has decided not to release the data on a public repository. The dataset is under the custodianship of the Research Administration at icddr,b. A de-identified analytical data set will be made available upon requests directed to the Research Administration of the icddr,b, at info@icddrb. org. Only after approval of a proposal data can be shared through a secure online platform. Approval of the proposal will be subject to sci-entific review by the institutional review board at icddr,b. Data sharing will also be subject to the published data access rules of the icddr. b. The requestor will need to sign a standard data access agreement required by the icddr,b.

**Funding:** Funding for this study was provided through a grant from the Eunice Kennedy Shriver National Institute of Child Health and Human Development of the US National Institutes of Health (1R21HD109819 to DTL, EJN, and AIK). The funders had no role in study design, data collection and analysis, decision to publish, or preparation of the manuscript.

**Competing interests:** The authors have declared that no competing interests exist.

providers. The findings highlight the multi-level drivers for inappropriate antibiotic use. Effective interventions should enhance clinical training, address economic incentives, promote community education, and establish clear regulatory frameworks. Future strategies should consider the interconnected nature of these influences rather than targeting isolated levels.

## Introduction

Antibiotic resistance has been recognized as one of the most persistent global public health challenges by the World Health Organization [1]. In 2019, it was estimated that antibiotic resistance was the direct cause of 1.2 million deaths worldwide and was a contributing factor to nearly 5 million more deaths [2]. Antibiotic resistance poses a particularly heavy burden in low- and middle-income countries (LMICs) and is expected to increase significantly, with estimates that antibiotic resistance will be a contributor to approximately 8 million deaths by the year 2050 [3]. The primary factor contributing to antibiotic resistance and the development and spread of antibiotic-resistant pathogens is the rise in the misuse and overuse of antibiotics over the past two decades [4].

In LMICs, antibiotics are often dispensed by informal healthcare providers, who fill an important gap in the shortage of formally trained healthcare professionals [5]. In the context of rural Bangladesh, "village doctors" are informal healthcare providers who play a central role in primary care, particularly in underserved areas. They often operate small, private clinics and/or medicine shops where they both recommend and directly provide medications, including antibiotics, to patients. Village doctors have no formal medical qualifications; often learning through apprenticeship, short training courses, or self-study. They differ from drug shop dispensers, who mainly sell medi-cations without providing clinical consultation or diagnosis. This group is an important target of antibiotic stewardship efforts by public health authorities, antimicrobial resis-tance (AMR) control programs, and the broader health sector in Bangladesh and sim-ilar low- and middle-income countries [2,3]. A study conducted in Bangladesh found that only 29% of antibiotics were dispensed through prescriptions by qualified doctors (MBBS), while 63% of antibiotics were dispensed directly by informal practitioners [6]. The study also revealed that antibiotics were most commonly used by children aged 0–15 years (35%), followed by individuals over 60 years old (23%) [6].

The use of antibiotics for the treatment of children with diarrheal disease is nota-bly high in LMICs [7]. Diarrheal diseases are a significant cause of illness and death in children worldwide, with the majority of cases and deaths occurring in LMICs [8]. Diarrheal diseases disproportionately affect impoverished communities and remain the second leading cause of death among children under five years of age [8]. Currently, decisions for using antibiotics and laboratory testing for acute diarrhea in children are mostly empiric. Although the World Health Organization (WHO) Inte-grated Management of Childhood Illness (IMCI) guidelines recommend antibiotics only for cases of bloody diarrhea or suspected cholera [9], inappropriate antibiotic use remains widespread.

Globally, both formally trained physicians and informal providers frequently prescribe antibiotics for non-bloody, non-severe diarrhea, driven by diagnostic uncertainty, perceived caregiver expectations, and limited time for counseling [9–13]. Studies from South and Southeast Asia, sub-Saharan Africa, and Latin America consistently report high rates of antibiotic recommendation or dispensing for uncomplicated pediatric diarrhea in both formal and informal sectors, with substantial between-country variation but a shared pattern of overuse [14–17]. In a recent survey of sites in LMICs, antibiotics are given to up to 70% of children without bloody diarrhea, even in countries where cholera is not known to be endemic [17].

In Bangladesh, the misuse and overuse of antibiotics are common, particularly through informal providers and over-the-counter sales without prescriptions [10,18–20]. This practice often involves broad-spectrum antibiotics for minor, viral illnesses, driven by caregiver demand and pharmaceutical marketing [10,21–23]. Moreover, evidence from Bangladesh indicates that inappropriate antibiotic prescribing for acute pediatric diarrhea also occurs within formal facilities, despite guideline recommendations emphasizing that in most cases treatment should be limited to Oral Rehydration Solution (ORS) and zinc supplementation [24,25].

In Bangladesh, informal healthcare providers, called "village doctors," are often the first point of contact for children with diarrhea, and these providers frequently dispense antibiotics for diarrheal illnesses, especially in rural areas [26,27]. Village doctors practice allopathic medicine with limited training and provide medical services to nearly two-thirds of the Bangladesh population. A study of village doctors in peri-urban Bangladesh found that 100% of patients seeking treatment for diarrhea from a village doctor received antibiotics, compared to 30% of those who consulted a community health worker (government-supported lay health workers with standardized training who provide basic health services and referrals but do not independently dispense medications) [28]. Most village doctors have limited knowledge about antibiotic use, resulting in varied and broad antibiotic dispensing practices that may have unintended harmful consequences at both individual and population levels [2].

Understanding antibiotic use in pediatric diarrhea treatment by informal providers is essential, particularly in settings where they serve as the primary source of care. Since informal providers often operate outside of formal regulatory frameworks, their antibiotic prescribing behaviors can be shaped by various factors. Qualitative methods such as in-depth interviews, focus group discussions and ethnographic observation are well suited for understanding how social factors contribute to the inappropriate use of antibiotics and barriers to guideline adherence [5,29]. These insights are crucial for designing context-specific antibiotic stewardship programs, strategies, and interventions. The goal of this study was to qualitatively examine the factors that influence of antibiotic dispensing for pediatric diarrhea among village doctors in rural Bangladesh. The results of the study will improve our understanding of the practices of informal healthcare providers regarding antibiotic use for pediatric diarrhea in rural Bangladesh and will help to inform interventions that promote responsible antibiotic use among these providers.

## Materials and methods

### Study design

This study is part of a larger project aimed at developing and testing a mobile health (mHealth) tool that supports village doctors in Bangladesh to use antibiotics responsibly [30]. The broader initiative includes mapping village doctors, conducting formative qualitative research to understand providers' behavior, co-designing the tool with users, pilot testing, and evaluating its impact.

As in the formative research phase, this study utilized a qualitative descriptive design to examine the factors that influence the provision of antibiotics among village doctors in rural Bangladesh. This approach enables researchers to gather comprehensive accounts of participants' lived experiences and viewpoints, which are essential to investigating complex behavioral patterns within specific contextual settings [31]. Through this methodology, we were able to explore the underlying motivations, beliefs, and situational factors that drive antibiotic provision decisions across different ecological levels.

## Study site and population

The study was conducted in *Sitakunda Upazila*, a sub-district of *Chattogram* District in the southeastern part of Bangladesh. *Sitakunda Upazila* is approximately 500 square kilometers of land area and has a population of around 497,396 [32]. The literacy average rate in *Sitakunda* was about 54%. The primary occupations of the local population are in the service sector (28.76%), agriculture (24.12%), and commerce (21.53%). The hospitals and healthcare centers in *Sitakunda* include Union Health Centre and community clinics at the wards [32]. *Sitakunda* was selected as a study site because it has high pediatric diarrhea burden [33]. The area has a large number of informal healthcare providers in practice in areas with a broad range of healthcare access and economic conditions. Our team knows the area well and has built strong, established working relationships with members of the community through previous research.

In the first stage of the broader initiative, using a 'snowball' sampling method, 411 village doctors were identified and mapped across 10 unions in Sitakunda, and 371 agreed to complete a structured survey, which has been reported elsewhere [34]. From this mapping, four unions were selected for participation in the qualitative interviews based on the following criteria: (i) accessibility:- areas that could be safely reached by the research team within reasonable travel time; (ii) density of village doctors:- unions with a sufficient number of active village doctors to enable adequate participant recruitment; and (iii) healthcare needs:- areas where village doctors are mostly popular as primary source of medical care for residents. From these four unions, we used convenience sampling to approach 20 village doctors to participate in the qualitative study. We targeted recruiting 20 village doctors because prior methodological work suggests 10–17 in-depth interviews within a relatively homogeneous group are typically sufficient to reach saturation and identify major patterns [35,36]. Moreover, we interviewed village doctors only, as our research focused specifically on the perspectives and practices of village doctors since they are a first point of care in rural settings.

From the 20 village doctors who were approached, 18 agreed to participate and completed in-depth interviews. The remaining two declined due to time constraints.

## Procedures

In-depth interviews (IDIs) were conducted by a team of three Bangladeshi anthropologists who were trained in qualitative research, familiar with the study objectives, and fluent in Bengali, the local language. We used in-depth interviews rather than other qualitative methods like focus group discussions due to the specific aim and context of our study. Individual interviews allow us to dive deeper into each participant's thoughts and experiences about antibiotic prescribing practices. This was especially important given the sensitivity of the possible misuse of antibiotics. By discussing one-on-one, participants could share their opinions freely, without the pressure of a group setting [37,38]. IDIs were conducted in May 2023 at locations convenient for the participants, typically their chambers or drug shops, to ensure a comfortable and familiar environment. Each interview lasted approximately 40–60 minutes and was audio-recorded with the participant's consent.

The team used a semi-structured interview guide (Supplement 1), developed in English and translated into Bengali. The guide covered key topics, including the role of village doctors in treating pediatric diarrhea, their knowledge of antibiotic stewardship, attitudes toward antibiotics, and practices regarding antibiotic dispensing behavior for pediatric diarrheal illness. All interviews were conducted in Bengali, audio recorded, transcribed verbatim, and then translated into English. The same three anthropologists who conducted the interviews were primarily responsible for transcription and translation, while all three team members collaboratively coded and analyzed the data to ensure consistency and rigor in interpretation.

## Analysis

We conducted an inductive thematic analysis to understand antibiotic dispensing among village doctors in rural Bangladesh. Village doctors both recommend and directly supply medications to patients outside the formal regulatory

framework. For this study, "antibiotic provision" refers to this combined process of clinical decision-making and medication supply, which differs from the separated prescribing and dispensing functions in formal healthcare settings.

Thematic analysis followed Braun and Clarke's approach [39], allowing themes to emerge directly from the interview data without imposing a pre-existing framework. The research team in Bangladesh reviewed all the interview transcripts, manually coded them for concepts related to antibiotic provision, and looked for relationships between codes, and grouped codes to synthesize the emerging themes [40].

After the initial coding and theme development, we mapped the inductively derived themes onto the five domains of the Social Ecological Framework (SEF) [41] that emerged as factors of antibiotic dispensing behavior across five interconnected levels: (1) Individual level, (2) Interpersonal level, (3) Organizational level, (4) Community level, and (5) Policy or macro level. This two-step process enabled us to remain grounded in participants' perspectives while also situating our findings within a widely recognized model for understanding multi-level influences on health behaviors. By organizing the results according to the SEF, we systematically examined the various factors influencing antibiotic provision practices for pediatric diarrhea among village doctors in rural Bangladesh, offering a comprehensive understanding of the issue within its broader socio-ecological context.

### Ethical approval

The study protocol was approved by the Ethical Review Committee of the International Centre for Diarrhoeal Diseases Research, Bangladesh (icddr,b), and the Institutional Review Board of the University of Utah, USA. The investigators explained the study to each participant and obtained informed written consent before conducting the interviews.

## Results

All 18 participants were male, aged between 28 and 63 years (mean: 41 years), and resided in the study area. Each had provided treatment for pediatric diarrhea for at least five years (Range: 5–41 years, Mean:19 years). Additional demographic and practice information for village doctors, including education, training, knowledge sources, and income, has been published elsewhere [34].

Our analysis revealed that antibiotic provision for pediatric diarrhea by village doctors in rural Bangladesh is influenced by a range of interconnected factors that span several levels of the social and healthcare environment (Fig 1). When managing cases of pediatric diarrhea, decisions about antibiotics are shaped by a combination of village doctors' clinical understanding and beliefs, expectations and demands from patients and pharmaceutical representatives, practical and financial considerations within their practice, prevailing community attitudes toward illness and treatment, and the broader context of limited regulatory oversight. These influences interact in complex ways, collectively shaping how antibiotics are provided in these settings.

### Individual level

At the individual level, several factors shaped village doctors' decisions to provide antibiotics for pediatric diarrhea. These included their clinical knowledge and training, personal beliefs about antibiotics, and approaches to clinical decision-making.

All participants reported having government-accredited training in medical practices, along with various short-term government and non-government training focused on the management of diarrheal diseases. However, this training did not necessarily translate into evidence-based clinical practice. Although clinical guidelines do not recommend providing antibiotics for most cases of diarrhea, most village doctors indicated that they routinely provided antibiotics to patients based on a belief that antibiotics are the best and quickest way to recover from illness.

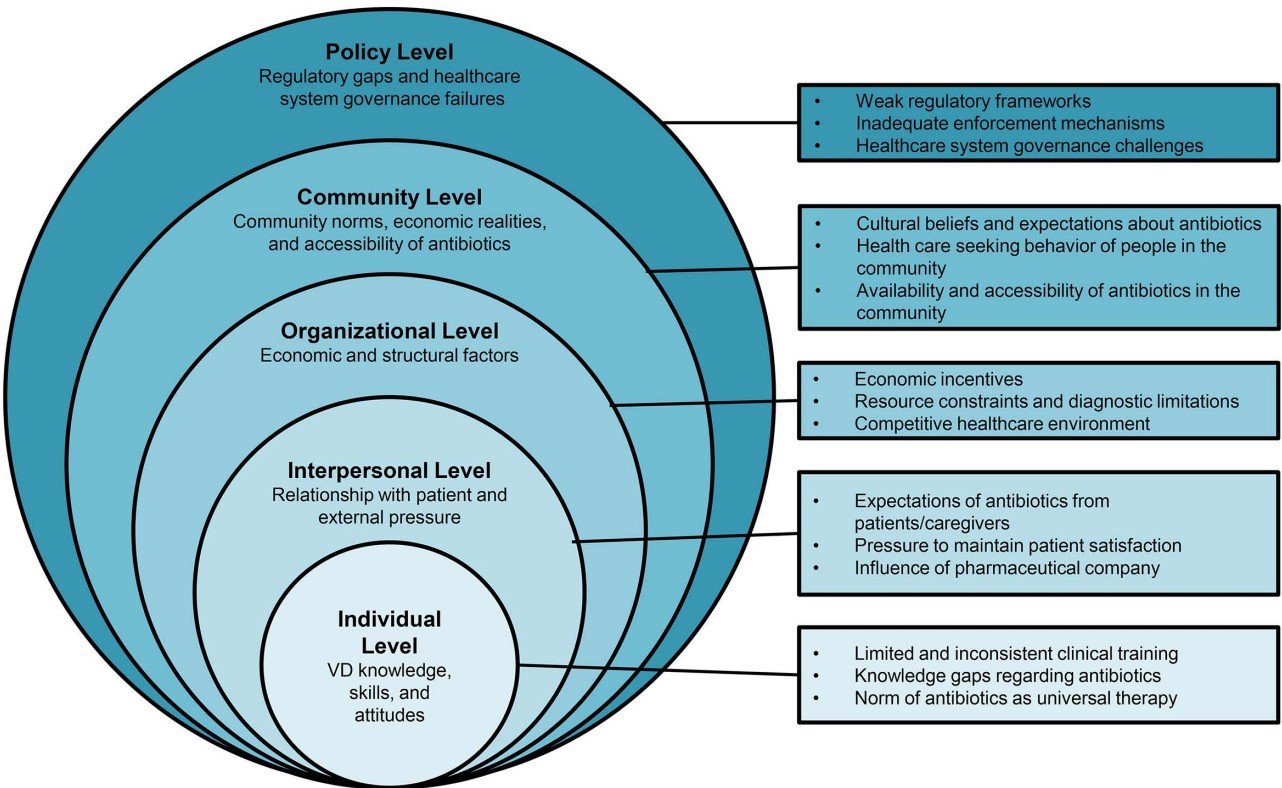

**Fig 1. Summary of themes across social ecological framework for antibiotic prescribing behavior.**

*"Antibiotics make patients recover faster, or when you have diarrhea, it works faster like magic. This is why antibiotics are being used without hesitation." (Village doctor with 21 years of practice)*

Knowledge gaps and misconceptions regarding antibiotics and antibiotic resistance were common. Many participants had limited understanding of what constitutes an antibiotic and demonstrated little awareness of antibiotic resistance as a public health concern. For example, they described giving patients metronidazole as a first line treatment but did not understand that it was an antibiotic.

*"I provide ORS, metronidazole, and probiotics as the first line of medications. If the condition does not improve, then antibiotics are provided."(Village doctor with 21 years of practice)*

Misconceptions about antibiotic spectrum were also observed. While many village doctors knew antibiotics are effective against bacteria, some believed that antibiotics could treat all pathogens, including viruses.

*"Antibiotics mainly work against any germs in our body. They kill the viruses that enter our bodies."* (Village doctor with 20 years of practice)

Few participants were familiar with the concept of antibiotic resistance. Although many had heard of it, none understood its precise definition or could explain the long-term consequences of antibiotic overuse or misuse. Moreover, most didn't perceive resistance as relevant to their daily practice or as a priority compared to immediate patient satisfaction and symptom relief.

In terms of clinical decision-making, the years of experience of village doctors appeared to play a role in antibiotic dispensing practices, with more junior village doctors being more likely to prescribe antibiotics for childhood diarrhea than senior village doctors. Some participants mentioned that senior practitioners may feel more confident in their clinical decision making while junior practitioners are more likely to have lower self-confidence in their clinical judgement. This lack of confidence may push them to prescribe antibiotics as a precaution to avoid potential complications, even when not strictly necessary or clinically indicated.

*"Those who are inexperienced, or those who are new, are still learning by working in a pharmacy (drug shop). They may not have any training, but they think of themselves as doctor; these junior people usually dispense more antibiotics."* (Village doctor with 21 years of practice)

### Interpersonal level

At the interpersonal level, village doctors' practices related to antibiotic provision were shaped by the expectations and demands of patients and caregivers, as well as the promotional activities of pharmaceutical companies.

Most participants explained that patients and their caregivers often expect and demand "powerful medicines" to ensure a quick recovery for their children. This expectation often centers on antibiotics, which are widely perceived as the most effective and rapid solution for pediatric diarrhea.

*"I think people have a tendency to take antibiotics because they believe antibiotics are powerful drugs for quick recovery."* (Village doctor with 21 years of practice)

*"Antibiotics help patients recover more quickly, especially during diarrhea. As they (caregivers/patient) asked us to provide antibiotics, we generally provide without any hesitation to make our customers happy."* (Village doctor with 21 years of practice)

All participants reported that parents typically take their child to a village doctor expecting to receive medications. In some cases, parents may even visit the village doctor without the pediatric patient to consult with the doctor and request specific medication for the sick child. Participants explained that client satisfaction with village doctors' services is largely dependent on the receipt of antibiotics, leading participants to feel pressured to provide antibiotics, even if not clinically necessary.

*"It is very common that sometimes parents come with their sick children and tell us that they saw a child with diarrhea in their neighborhood recover quickly with [Azithromycin/ Cefixime]. So, they believe their baby will have a rapid cure and thus demand [Azithromycin/Cefixime]. Sometimes, they come with old packets or syrup bottles."* (Village doctor with 15 years of practice)

In addition to client expectations, participants also spoke about the influence of pharmaceutical companies on their antibiotic providing practices. Pharmaceutical companies frequently organize or support short-term training on management of various diseases, including diarrhea. Participants noted that these programs often emphasize selling their products, rather than providing comprehensive education on the rational use of antibiotics.

*"Pharmaceutical companies sometimes organize seminars to promote their medications, not arrange programs or training that could help us learn new things."* (Village doctor with 9 years of practice)

### Organizational level

At the organizational level, several factors influenced the antibiotic provision practices of village doctors, including economic incentives, resource constraints, and a competitive healthcare environment.

The economic structure of village doctors' practices contributes to the provision of antibiotics. Village doctors can earn income through both consultation fees and the sale of medications. For some, the sale of medicines was their sole source of income, as they didn't charge for consultation. This dual role of serving as both healthcare provider and medicine seller creates inherent conflicts of interest that could influence their recommendations to patients. Some participants said they did not charge any consultation fees and instead relied solely on selling medications directly to patients for their income. This business model necessitates maintaining a stock of medications and managing operational costs, which can create pressure to sell the available stock of medication. Additionally, pharmaceutical companies offer financial and non-financial incentives to village doctors who promote their products. Since antibiotics are typically priced higher than other medications, village doctors earn larger commissions and profit margins for selling antibiotics.

*"A well-known brand's tablet has a lower profit margin. However, if you sell antibiotics from an unknown brand, you will achieve a better profit margin, even though those medicines may be less effective."* (Village doctor with 20 years of practice)

*"If I sell ten boxes of antibiotics, they often reward me with two additional boxes as a gift. Sometimes they even provide gifts like utensils or fans."* (Village doctor with 21 years of practice)

Resource constraints were another organizational challenge. Most village doctors lacked access to laboratory facilities and advanced diagnostic equipment. This limitation hinders their ability to make evidence-based prescribing decisions, potentially leading them to adopt overly cautious prescribing practices. Additionally, the competitive nature of the healthcare environment in rural areas, where multiple providers vie for patients, further reinforced the tendency to prescribe antibiotics as a means of maintaining patient satisfaction and sustaining their business.

*"In many cases, particularly in rural or small-town areas, pharmacies and local doctors often prescribe unnecessary antibiotics primarily for business reasons. Their motivation is not related to treating diseases but rather to profit from selling antibiotics, which can be quite lucrative."* (Village doctor with 5 years of practice)

Since village doctors had limited access to laboratory diagnostics, they relied primarily on patient history and clinical presentation, including noticeable color and odor of stool samples. Although this approach is pragmatic given the constraints, it may result in over-use of antibiotics.

*"When someone suffers from diarrhea, and has a bad odor in stool, we consider it a bacterial infection and provide antibiotics to that patient."* (Village doctor with 27 years of practice)

## Community level

At the community level, cultural beliefs and expectations, healthcare seeking behaviors within the community, and antibiotic availability and accessibility play a role in shaping antibiotic provision practices. Community members believed that antibiotics are essential for a quick recovery from illness. Participants felt that many members of the community were unfamiliar with the term "antibiotic," but they recognized the names of specific medications. Those who were aware of antibiotics often viewed them as potent and highly effective treatments for a range of diseases, particularly diarrhea.

*"In Bangladesh, I don't think any patient is satisfied without taking any antibiotic, whether its diarrhea or cough and cold with fever or any other illness. That's what I have seen in my years of practice. Sometimes you can't be a good doctor if you don't give antibiotic."* (Village doctor with 5 years of practice)

Village doctors also explained that people in their community often changed village doctors if they didn't see improvement in a short time. This further reinforces the cultural expectations of trying to minimize recovery time by prescribing antibiotics instead of taking a "wait and see" approach.

The socio-economic conditions of rural communities create additional complexities. Many village doctors mentioned that most patients were from low-income families. Some doctors take the patient's economic situation into account when deciding which medications to sell to patients.

"*If doxycycline is sufficient for my patient, I won't prescribe ciprofloxacin. Doxycycline costs 2 takas, while ciprofloxacin costs 20 takas, so the patient would save 18 takas.*" (Village doctor with 17 years of practice)

Finally, participants reported that antibiotics were widely available from multiple venues in their community. The availability and promotion of antibiotics in the community sustains the expectation that antibiotics can and should be taken to treat all common ailments.

## Policy level

At the policy level, antibiotic provision among village doctors was influenced by weak regulatory frameworks, inadequate enforcement mechanisms, and broader governance challenges within Bangladesh's healthcare system.

The most significant policy level factor was the lack of stringent regulations governing antibiotic sales and distribution. This regulatory weakness is compounded by widespread lack of awareness about prescribing authority, that is, most participants were unaware that the right to prescribe antibiotics legally belongs to registered physicians only. Current policies provide limited oversight of antibiotic sales and distribution in rural areas, with minimal monitoring of prescribing patterns. Participants also described this lack of strict guidelines for antibiotic dispensing creates an environment where antibiotics could be purchased freely from unregulated outlets, including grocery stores and stationery shops, where untrained individuals dispensed medications without proper knowledge or qualifications. This unregulated environment is exacerbated by the widespread use of counterfeit certificates by pharmacy operators, undermining the credibility and safety of healthcare services.

"*Whether it is a betel leaf store or a stationery store, you can find medicines/drugs to some extent, particularly for diarrhea. Go to any stationery shop and tell them to give you something for diarrhea, then they will give you oral dehydration saline along with 'Metronidazole' certainly.*" (Village doctor with 5 years of practice)

The weak regulatory system is further undermined by inadequate enforcement efforts. Notably, none of the participants reported any legal penalties associated with selling antibiotics, highlighting a gap in enforcement that could warn practitioners about legal risks. Instead, village doctors expressed confidence in their training and management skills, asserting they are well-equipped to treat diarrheal diseases and believe they have the authority to provide antibiotics for this illness.

"*There are no real rules and regulations for selling and consumption of antibiotic, as a result of which the people of Bangladesh are suffering a lot.*" (Village doctor with 5 years of practice)

The policy environment also created conditions that positioned village doctors as essential healthcare providers in rural areas, despite their lack of formal recognition. Village doctors play a crucial role in providing healthcare in rural areas, where access to formal healthcare options is often limited and challenging. Caregivers of children with diarrhea often avoid visiting government hospitals due to challenges of distance, cost, long wait time and lack of trust. Given this, parents often prefer to take their child to a village doctor for more timely treatments and cost-effective services. Although these

village doctors are not officially recognized by the government, the community acknowledges them for their availability and the services they provide. This recognition allows them considerable freedom to use antibiotics for treating pediatric diarrhea.

*"Most families are not willing to go to the hospital. They like to get initial treatment from their family doctor (village doctor). As they are our old patients and known people, they first come to us. Another reason is the money. Some people think about the distance to a hospital."* (Village doctor with 21 years of practice)

## Discussion

Our study utilized the Social Ecological Framework to describe the complex factors that drive antibiotic dispensing among village doctors in rural Bangladesh. The results identified interconnected factors at individual, interpersonal, organizational, community, and policy levels, and point to the need for multi-level interventions to curb inappropriate antibiotic use among village doctors.

At the individual level, our findings reveal a significant gap in both knowledge and practice regarding the treatment of pediatric diarrhea. Village doctors showed limited understanding of antimicrobial resistance and the proper use of antibiotics. Although most of the participants had received some form of training in diarrhea management, their antibiotic dispensing practices did not align with standard clinical guidelines. The low educational levels of these village doctors may contribute to their inadequate understanding of when antibiotics are useful and the potential harm they may introduce for patients. Similar findings have been reported in studies from low- and middle-income countries, indicating that informal healthcare providers often lack a comprehensive understanding of rational and appropriate antibiotic use [42–44].

The interpersonal and organizational factors highlight how economic incentives and patient expectations affect the prescribing behaviors of village doctors. Village doctors experience pressures to sell antibiotics from multiple directions. Patients and the community expect antibiotics as the primary treatment, and the doctors must also meet their financial needs through the sale of medications. In this context, a village doctor's income is closely linked to medication sales, and incentives from pharmaceutical companies for selling antibiotics offer further financial gains. The combination of economic incentives, patient and community expectations, and the desire to retain clientele, creates a challenging environment for practicing rational antibiotic usage. Studies across South Asia have identified these outside pressures on informal healthcare providers to dispense antibiotics, and the disincentive for providers to limit antibiotic sales [5]. In particular, financial incentives from pharmaceutical companies are a highly influential factor driving unnecessary antibiotic use not only in Bangladesh [10], but also in Korea [45], India [46], and China [44].

Our study highlights how cultural beliefs and access to healthcare impact and shape antibiotic use at the community level. In rural areas, community trust in village doctors and barriers to accessing formal healthcare facilities make these local informal healthcare providers essential. The accessibility of village doctors, coupled with the belief that antibiotics are "stronger medicine," create the conditions for over usage of antibiotics. In settings like Bangladesh where populations can go directly to informal providers to get medications, even the best efforts for facility-based antibiotic stewardship can be undermined [47,48].

Despite Bangladesh's National Drug Policy, which prohibits the sale and distribution of antibiotics without a prescription from a registered physician [49], village doctors continue to dispense these medications. The lack of clear regulations governing the practice of village doctors, coupled with weak enforcement mechanisms and limited integration into the formal healthcare system, creates an environment where inappropriate antibiotic dispensing and use can flourish. This regulatory gap is particularly concerning given the essential role village doctors play in providing healthcare services in rural areas. This challenge is not unique to Bangladesh; similar regulatory challenges have been observed in other developing countries, highlighting the need for balanced policy approaches that recognize informal healthcare providers while promoting better antibiotic stewardship practices [27,50].

This study has several notable limitations. Firstly, it was conducted in only one *upazila* in the Chattogram district, which makes it challenging to generalize the findings to other regions. However, the study does provide compelling evidence regarding knowledge, attitudes, and practices related to antibiotic dispensing among village doctors that is relevant to other areas of the country. Secondly, the study included only 18 village doctor participants. While the sample was small, data saturation was carefully assessed and confirmed. Third, this study didn't include other stakeholders like patients, caregivers and drug sellers. Future studies would benefit from a more inclusive approach that incorporates feedback from these stakeholders, facilitating a holistic understanding of the antibiotic supply chain. Another important limitation is, this study did not examine how village doctors and formal healthcare providers interact or if there are any conflicts. Issues such as competition, referrals, respect and collaboration can create both barriers and facilitators to appropriate antibiotic use. Future research should explore these relationships to help create better strategies for addressing antibiotic resistance.

Despite these limitations, this study highlights several key factors that affect the behavior of village doctors in Bangladesh surrounding antibiotic usage for the treatment of pediatric diarrhea. Given the vital role of village doctors in rural areas of Bangladesh, it is crucial to implement interventions that encourage responsible use of antibiotics. This study indicates that a single intervention is unlikely to effectively improve the antibiotic dispensing practices of informal healthcare providers. Instead, a comprehensive approach that addresses multiple levels simultaneously is necessary. This approach could involve educating both healthcare providers and communities about the rational use of antibiotics and the consequences of their unnecessary use, and creating economic incentives for village doctors to reduce antibiotic use. Additionally, policy reforms and a reduction in the influence of pharmaceutical companies are essential. It is also crucial to enforce existing laws related to antibiotic usage. Ultimately, antibiotic stewardship is a global imperative that requires coordinated effort from multiple stakeholders to ensure that antibiotics remain effective in treating infections, safeguarding public health for generations to come.

## Acknowledgments

We are thankful to the study participants for their participation and the valuable information they provided for this study. icddr,b is grateful to the governments of Bangladesh and Canada for providing unrestricted support.

**Disclosure:** The findings and conclusions in this report are those of the authors and do not necessarily reflect the opinions of the institutions with which the authors are affiliated.

## Author contributions

**Conceptualization:** Debashish Biswas, Melissa H. Watt, Md. Taufiqul Islam, Eric J. Nelson, Firdausi Qadri, Daniel Leung, Ashraful Islam Khan.

**Data curation:** Debashish Biswas, Melissa H. Watt, Aparna Mangadu, Olivia R. Hanson.

**Formal analysis:** Debashish Biswas, Jyoti Bhushan Das, Mohammad Saeed Munim, Ridwan Mostofa Shihab.

**Funding acquisition:** Eric J. Nelson, Daniel Leung, Ashraful Islam Khan.

**Investigation:** Debashish Biswas, Jyoti Bhushan Das, Mohammad Saeed Munim, Ridwan Mostofa Shihab, Md. Taufiqul Islam.

**Methodology:** Debashish Biswas, Melissa H. Watt, Olivia R. Hanson, Daniel Leung.

**Project administration:** Debashish Biswas, Zahid Hasan Khan, Mohammad Ashraful Amin, Ishtiakul Islam Khan, Md. Taufiqul Islam.

**Resources:** Firdausi Qadri, Daniel Leung, Ashraful Islam Khan.

**Supervision:** Debashish Biswas, Ashraful Islam Khan.

**Validation:** Debashish Biswas, Melissa H. Watt, Eric J. Nelson, Daniel Leung.

**Visualization:** Debashish Biswas, Aparna Mangadu, Jyoti Bhushan Das, Mohammad Saeed Munim, Ridwan Mostofa Shihab.

**Writing – original draft:** Debashish Biswas.

**Writing – review & editing:** Melissa H. Watt, Aparna Mangadu, Jyoti Bhushan Das, Mohammad Saeed Munim, Ridwan Mostofa Shihab, Zahid Hasan Khan, Mohammad Ashraful Amin, Ishtiakul Islam Khan, Md. Taufiqul Islam, Olivia R. Hanson, Eric J. Nelson, Firdausi Qadri, Daniel Leung, Ashraful Islam Khan.

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
