## [Decision Letter · Decision Letter 0]

27 Aug 2025

Antibiotic dispensing behaviors for pediatric diarrhea: A qualitative study of informal healthcare providers in rural Bangladesh

PLOS ONE

Dear Dr.  Biswas,

Thank you for submitting your manuscript to PLOS ONE. After careful consideration, we feel that it has merit but does not fully meet PLOS ONE’s publication criteria as it currently stands. Therefore, we invite you to submit a revised version of the manuscript that addresses the points raised during the review process.

https://journals.plos.org/plosone/s/submission-guidelines#loc-laboratory-protocols . Additionally, PLOS ONE offers an option for publishing peer-reviewed Lab Protocol articles, which describe protocols hosted on protocols.io. Read more information on sharing protocols at https://plos.org/protocols?utm_medium=editorial-email&utm_source=authorletters&utm_campaign=protocols .

We look forward to receiving your revised manuscript.

Kind regards,

Md. Golam Dostogir Harun, MSS, MPH

Academic Editor

PLOS ONE

Journal Requirements:

“Funding for this study was provided through a grant from the Eunice Kennedy Shriver National Institute of Child Health and Human Development of the US National Institutes of Health (1R21HD109819 to DTL, EJN, and AIK).”

4. Please note that funding information should not appear in any section or other areas of your manuscript. We will only publish funding information present in the Funding Statement section of the online submission form. Please remove any funding-related text from the manuscript.

Additional Editor Comments:

This manuscript is well-written and significant. Both reviewers provide critical feedback on it. Please address their comments and submit a revised version.

Reviewers' comments:

Reviewer's Responses to Questions

**Comments to the Author**

1. Is the manuscript technically sound, and do the data support the conclusions?

Reviewer #1: Partly

Reviewer #2: Yes

2. Has the statistical analysis been performed appropriately and rigorously?

Reviewer #1: N/A

Reviewer #2: N/A

3. Have the authors made all data underlying the findings in their manuscript fully available?

Reviewer #1: Yes

Reviewer #2: Yes

4. Is the manuscript presented in an intelligible fashion and written in standard English?

Reviewer #1: Yes

Reviewer #2: Yes

Reviewer #1: Introduction:

1.Please write the practice of antibiotic use for treating diarrhoea by physicians and informal providers globally

2.Please provide clinical explanation of antibiotic use in pediatric diarrhoea treatment

3.Please include misuse and overuse of antibiotics in Bangladesh

4.Please write the practice of antibiotic use for treating diarrhoea by physicians in Bangladesh

5.Please expand the last paragraph that includes the necessity of exploring antibiotic use in pediatric diarrhoea treatment by informal providers, and necessity of qualitative exploration (with proper citation)

Method:

6.“Sitakunda was selected as a study site because of our team’s familiarity with the area and established working relationships with members of the community”-this should not be the reasons of conducting study in a particular area. Please explain other reasons that convince us to choose this area.

7.“This study is part of a larger initiative focused on developing and evaluating a mobile health tool to improve antibiotic stewardship among village doctors in Bangladesh- please mention this at the begging of the method and explain the relevancy of the study with the larger initiative.

8. We need to know the context of the study areas such as healthcare facilities, accessibility of the healthcare facilities, literacy of the people, economic status, education, profession etc

9.Please explain why you selected only 20 village doctors for IDI among large population (371 village doctors). Why you did not use other qualitative tools such as FGD, observations and case study.Why did not you include patients and their parents and drug sellers in the study?

Result:

10.You mentioned 20 village doctors in the methods, but now you mention 18. Please make clear in the manuscript.

11.Demographic information is not sufficient. You need to include education, training or source of knowledge, income from the treatment etc.

12.In Bangladesh, village doctors provide treatment and provide antibiotic for both human and animal. Do you have any findings related to that?

13.It is necessary to know the knowledge and perception of negative impact of using antibiotic and antibiotic resistance from village doctors. Please write details about that with quotation.

14.Do the village doctors have any conflict with formal healthcare providers? Please explain.

Reviewer #2: Thanks for developing a manuscript titled "Antibiotic dispensing behaviors for pediatric diarrhea: A qualitative study of informal healthcare providers in rural Bangladesh", which is crucial for the sector. Overall, the paper is well written. However, please find some specific areas where we have the opportunity to improve:

Line 82: Are they village doctors? or medicine dispensers? What is the qualification of these village doctors? Please add this somewhere at the beginning of the paper. Also, you may clarify for the reader that village doctors and shop dispenser are different.

Line 84: Private clinic and medicine shops are not same, so please edit "or", maybe "and" is appropriate here

Line 86: Important target for whom? Government? The entire sector? One health?

Line 111: Community health worker introduced here for the first time, may be better to add one sentence on them earlier to this when you mention village doctors. i. e. Who are these community health workers? Why are they more qualified than village doctors?

Lines 117-119: Maybe it's better to explain the research gap here first, and then describe how qualitative research can contribute.

137: It seems that this is not a stronger justification for selecting the field site. Maybe the next paragraph is talking about the justification of the field sites, which seems good.

Figure: The quality of the figure should be better; it's distorted now.

**Do you want your identity to be public for this peer review?** For information about this choice, including consent withdrawal, please see our Privacy Policy

Reviewer #1: **Yes: ** S M Murshid Hasan

Reviewer #2: **Yes: ** Mahbub-Ul Alam

---

## [Author Response · Author response to Decision Letter 1]

15 Oct 2025

We have uploaded a separate document containing all of our responses to the reviewers.

---

## [Editor Report · Decision Letter 1]

5 Nov 2025

Dear Dr. Biswas,

Thank you for submitting your manuscript to PLOS ONE. After careful consideration, we feel that it has merit but does not fully meet PLOS ONE’s publication criteria as it currently stands. Therefore, we invite you to submit a revised version of the manuscript that addresses the points raised during the review process.

We look forward to receiving your revised manuscript.

Kind regards,

Md. Golam Dostogir Harun, MSS, MPH

Academic Editor

PLOS ONE

Journal Requirements:

Additional Editor Comments:

Please revise the reviewers comments and resubmit the revise manuscript

---

## [Author Response · Author response to Decision Letter 2]

30 Nov 2025

No new feedback was provided and that all previous reviewer comments and editorial concerns have been addressed.

---

## [Editor Report · Decision Letter 2]

2 Dec 2025

Antibiotic dispensing behaviors for pediatric diarrhea: A qualitative study of informal healthcare providers in rural Bangladesh

PONE-D-25-34609R2

Dear Dr.Debashish,

We’re pleased to inform you that your manuscript has been judged scientifically suitable for publication and will be formally accepted for publication once it meets all outstanding technical requirements.

Kind regards,

Md. Golam Dostogir Harun, MSS, MPH

Academic Editor

PLOS ONE

Additional Editor Comments (optional):

Dear Dr. Debashish,

Thank you for addressing the reviewers' comments point by point and for revising the manuscript. Congratulations! Your manuscript has now been accepted for publication.
---

## [Editor Report · Acceptance letter]

PONE-D-25-34609R2

PLOS One

Dear Dr. Biswas,

I'm pleased to inform you that your manuscript has been deemed suitable for publication in PLOS One. Congratulations! Your manuscript is now being handed over to our production team.

Kind regards,

on behalf of

Dr. Md. Golam Dostogir Harun

Academic Editor

PLOS One